# One-shot learning: From domain knowledge to action models

## Abstract

Most approaches to learning action planning models heavily rely on a significantly large volume of training samples or plan observations. In this paper, we adopt a different approach based on deductive learning from domain-specific knowledge, specifically from logic formulae that specify constraints about the possible states of a given domain. The minimal input observability required by our approach is a single example composed of a full initial state and a partial goal state. We will show that exploiting specific domain knowledge enables to constrain the space of possible action models as well as to complete partial observations, both of which turn out helpful to learn good-quality action models.

## Introduction

The learning of action models in planning has been typically addressed with inductive learning data-intensive approaches. From the pioneer learning system ARMS (Yang *et al.* 2007) to more recent ones (Mourão *et al.* 2012; Zhuo and Kambhampati 2013; Kucera and Barták 2018), all of them require thousands of plan observations or training samples, i.e., sequences of actions as evidence of the execution of an observed agent, to obtain and validate an action model. These approaches return the statistically significant model that best explains the plan observations by minimizing some error metric. A model explains an observation if a plan containing the observed actions is computable with the model and the states induced by this plan also include the possibly partially observed states. The limitation of posing model learning and validation as optimization tasks over a set of observations is that it neither guarantees completeness (the model may not explain all the observations) nor correctness (the states induced by the execution of the plan generated with the model may contain contradictory information).

Differently, other approaches rely on symbolic-via learning. The Simultaneous Learning and Filtering (SLAF) approach (Amir and Chang 2008) exploits logical inference and builds a complete explanation through a CNF formula that represents the initial belief state, and a plan observation. The formula is updated with every action and state of the observation, thus representing all possible transition relations consistent with it. SLAF extracts all satisfying models of the learned formula with a SAT solver although the algorithm cannot effectively learn the preconditions of actions. A more recent approach addresses the learning of action models from plan observations as a planning task which searches the space of all possible action models (Aineto *et al.* 2018). A plan here is conceived as a series of steps that determine the preconditions and effects of the action models plus other steps that validate the formed actions in the observations. The advantage of this approach is that it only requires input samples of about a total of 50 actions.

This paper studies the impact of using mixed input data, i.e, automatically-collected plan observations and human-encoded domain-specific knowledge, in the learning of action models. Particularly, we aim to stress the extreme case of having a single observation sample and answer the question to whether the lack of training samples can be overcome with the supply of domain knowledge. The question is motivated by (a) the assumption that obtaining enough training observations is often difficult and costly, if not impossible in some domains (Zhuo 2015); (b) the fact that although the physics of the real-world domain being modeled are unknown, the user may know certain pieces of knowledge about the domain; and (c) the desire for correct action models that are usable beyond their fitness to a set of testing observations. To this end, we opted for checking our hypothesis in the framework proposed in (Aineto *et al.* 2018) since this planning-based satisfiability approach allows us to configure additional constraints in the compilation scheme, it is able to work under a minimal set of observations and uses an off-the-shelf planner[1]. Ultimately, we aim to compare the informational power of domain observations (information quantity) with the representational power of domain-specific knowledge (information quality). Complementarily, we restrict our attention to solely observations over fluents as in many applications the actual actions of an agent may not be observable (Sohrabi *et al.* 2016).

Next section summarizes basic planning concepts and outlines the baseline learning approach (Aineto *et al.* 2018). Then we formalize our one-shot learning task with domain

---

[1]We thank authors for providing us with the source files of their learning system.

knowledge and subsequently we explain the task-solving process. Section 5 presents the experimental evaluation and last section concludes.

## Background

We denote as $F$ (fluents) the set of propositional state variables. A partial assignment of values to fluents is represented by $L$ (literals). We adopt the *open world assumption* (what is not known to be true in a state is unknown) to implicitly represent the unobserved literals of a state. Hence, a state $s$ includes positive literals ($f$) and negative literals ($\neg f$) and it is defined as a full assignment of values to fluents; $|s| = |F|$. We use $\mathcal{L}(F)$ to denote the set of all literal sets on $F$; i.e. all partial assignments of values to fluents.

A *planning action* $a$ has a precondition list $\mathsf{pre}(a) \in \mathcal{L}(F)$ and a effect list $\mathsf{eff}(a) \in \mathcal{L}(F)$. The semantics of an action $a$ is specified with two functions: $\rho(s, a)$ denotes whether $a$ is *applicable* in a state $s$ and $\theta(s, a)$ denotes the *successor state* that results from applying $a$ in a state $s$. Then, $\rho(s, a)$ holds iff $\mathsf{pre}(a) \subseteq s$, i.e. if its preconditions hold in $s$. The result of executing an applicable action $a$ in a state $s$ is a new state $\theta(s, a) = \{s \setminus \neg\mathsf{eff}(a) \cup \mathsf{eff}(a)\}$, where $\neg\mathsf{eff}(a)$ is the complement of $\mathsf{eff}(a)$, which is subtracted from $s$ so as to ensure that $\theta(s, a)$ remains a well-defined state. The subset of effects of an action $a$ that assign a positive value to a fluent is called *positive effects* and denoted by $\mathsf{eff}^+(a) \in \mathsf{eff}(a)$ while $\mathsf{eff}^-(a) \in \mathsf{eff}(a)$ denotes the *negative effects*.

A *planning problem* is a tuple $P = \langle F, A, I, G \rangle$, where $I$ is the initial state and $G \in \mathcal{L}(F)$ is the set of goal conditions over the state variables. A *plan* $\pi$ is an action sequence $\pi = \langle a_1, \ldots, a_n \rangle$, with $|\pi| = n$ denoting its *plan length*. The execution of $\pi$ in $I$ induces a *trajectory* $\langle s_0, a_1, s_1, \ldots, a_n, s_n \rangle$ such that $s_0 = I$ and, for each $1 \le i \le n$, it holds $\rho(s_{i-1}, a_i)$ and $s_i = \theta(s_{i-1}, a_i)$. A plan $\pi$ solves $P$ iff the induced trajectory reaches a final state $s_n$ such that $G \subseteq s_n$.

The baseline learning approach our proposal draws upon uses *actions with conditional effects* (Aineto *et al.* 2018). The conditional effects of an action $a_c$ is composed of two sets of literals: $C \in \mathcal{L}(F)$, the *condition*, and $E \in \mathcal{L}(F)$, the *effect*. The *triggered effects* resulting from the action application (conditional effects whose conditions hold in $s$) is defined as $\mathsf{eff}_c(s, a) = \bigcup_{C \triangleright E \in \mathsf{cond}(a_c), C \subseteq s} E$.

### Learning action models as planning

The approach for learning STRIPS action models presented in (Aineto *et al.* 2018), which we will use as our baseline learning system (hereafter BLS, for short), is a compilation scheme that transforms the problem of learning the preconditions and effects of action models into a planning task $P'$. A STRIPS *action model* $\xi$ is defined as $\xi = \langle name(\xi), pars(\xi), pre(\xi), add(\xi), del(\xi) \rangle$, where $name(\xi)$ and parameters, $pars(\xi)$, define the header of $\xi$; and $pre(\xi)$, $del(\xi)$ and $add(\xi))$ are sets of fluents that represent the *preconditions*, *negative effects* and *positive effects*, respectively, of the actions induced from the action model $\xi$.

The BLS receives as input an empty domain model, which only contains the headers of the action models, and a set of observations of plan executions, and creates a propositional encoding of the planning task $P'$. Let $\Psi$ be the set of *predicates*[2] that shape the variables $F$. The set of propositions of $P'$ that can appear in $pre(\xi)$, $del(\xi)$ and $add(\xi)$ of a given $\xi$, denoted as $\mathcal{I}_{\xi, \Psi}$, are FOL interpretations of $\Psi$ over the parameters $pars(\xi)$. For instance, in a four-operator *blocksworld* (Slaney and Thiébaux 2001), the $\mathcal{I}_{\xi, \Psi}$ set contains five elements for the `pickup(`$v_1$`)` model, $\mathcal{I}_{pickup, \Psi}$=`{handempty, holding(`$v_1$`),clear(`$v_1$`),ontable(`$v_1$`), on(`$v_1, v_1$`)}` and eleven elements for the model of `stack(`$v_1, v_2$`)`, $\mathcal{I}_{stack, \Psi}$=`{handempty, holding(`$v_1$`), holding(`$v_2$`), clear(`$v_1$`),clear(`$v_2$`),ontable(`$v_1$`),ontable(`$v_2$`), on(`$v_1, v_1$`),on(`$v_1, v_2$`), on(`$v_2, v_1$`), on(`$v_2, v_2$`)}`. Hence, solving $P'$ consists in determining which elements of $\mathcal{I}_{\xi, \Psi}$ will shape the preconditions, positive and negative effects of each action model $\xi$.

The decision as to whether or not an element of $\mathcal{I}_{\xi, \Psi}$ will be part of $pre(\xi)$, $del(\xi)$ or $add(\xi)$ is given by the plan that solves $P'$. Specifically, two different sets of actions are included in the definition of $P'$: *insert actions*, which insert preconditions and effects on an action model; and *apply actions*, which validate the application of the learned action models in the input observations. Roughly speaking, in the *blocksworld* domain, the insert actions of a plan that solves $P'$ will look like (`insert_pre_stack_holding_v1`), (`insert_eff_stack_clear_v1`), (`insert_eff_stack_clear_v2`), where the second action denotes a positive effect and the third one a negative effect both to be inserted in the model of `stack`; and the second set of actions of the plan that solves $P'$ will be like (`apply_unstack blockB blockA`),(`validate_1`),(`apply_putdown blockB`),(`validate_2`), where the `validate` actions denote the points at which the states generated through the `apply` actions must be validated with the observations of plan executions.

In a nutshell, the output of the BLS compilation is a plan that completes the empty input domain model by specifying the preconditions and effects of each action model such that the validation of the completed model over the input observations is successful.

## *One-shot* learning task

The *one-shot* learning task to learn action models from *domain-specific knowledge* is defined as a tuple $\Lambda = \langle \mathcal{M}, \mathcal{O}, \Phi \rangle$, where:

- $\mathcal{M}$ is the *initial empty model* that contains only the header of each action model to be learned.

- $\mathcal{O}$ is a single learning example or plan observation; i.e. a sequence of (partially) observable states representing the evidence of the execution of an observed agent.

- $\Phi$ is a set of logic formulae that define *domain-specific knowledge*.

---

[2]The initial state of an observation is a full assignment of values to fluents, $|s_0| = |F|$, and so the predicates $\Psi$ are extractable from the observed state $s_0$.

| Combination | Meaning |
|---|---|
| $\neg pre\_\xi\_e \wedge \neg eff\_\xi\_e$ | $e$ belongs neither to the preconditions nor effects of $\xi$ $(e \notin pre(\xi) \wedge e \notin add(\xi) \wedge e \notin del(\xi))$ |
| $pre\_\xi\_e \wedge \neg eff\_\xi\_e$ | $e$ is only a precondition of $\xi$ $(e \in pre(\xi) \wedge e \notin add(\xi) \wedge e \notin del(\xi))$ |
| $\neg pre\_\xi\_e \wedge eff\_\xi\_e$ | $e$ is a positive effect of $\xi$ $(e \notin pre(\xi) \wedge e \in add(\xi) \wedge e \notin del(\xi))$ |
| $pre\_\xi\_e \wedge eff\_\xi\_e$ | $e$ is a negative effect of $\xi$ $(e \in pre(\xi) \wedge e \notin add(\xi) \wedge e \in add(\xi))$ |

Figure 1: Combinations of the fluent propositional encoding and their meaning

A *solution* to a learning task $\Lambda = \langle \mathcal{M}, \mathcal{O}, \Phi \rangle$ is a model $\mathcal{M}'$ s.t. there exists a plan computable with $\mathcal{M}'$ that is consistent with the headers of $\mathcal{M}$, the observed states of $\mathcal{O}$ and the given domain knowledge in $\Phi$.

## The space of STRIPS action models

We analyze here the solution space of a learning task $\Lambda = \langle \mathcal{M}, \mathcal{O}, \Phi \rangle$; i.e., the space of STRIPS action models. In principle, for a given action model $\xi$, any element of $\mathcal{I}_{\xi,\Psi}$ can potentially appear in $pre(\xi)$, $del(\xi)$ and $add(\xi)$. In practice, the actual space of possible STRIPS schemata is bounded by:

1. **Syntactic constraints**. The solution $\mathcal{M}'$ must be consistent with the STRIPS constraints: $del(\xi) \subseteq pre(\xi)$, $del(\xi) \cap add(\xi) = \emptyset$ and $pre(\xi) \cap add(\xi) = \emptyset$. *Typing constraints* would also be a type of syntactic constraint (McDermott *et al.* 1998).

2. **Observation constraints**. The solution $\mathcal{M}'$ must be consistent with these *semantic constraints* derived from the learning samples $\mathcal{O}$, which in our case is a single plan observation. Specifically, the states induced by the plan computable with $\mathcal{M}'$ must comprise the observed states of the sample, which further constrains the space of possible action models.

Considering only the syntactic constraints, the size of the space of possible STRIPS models is given by $2^{2 \times |\mathcal{I}_{\Psi,\xi}|}$ because one element in $\mathcal{I}_{\xi,\Psi}$ can appear both in the preconditions and effects of $\xi$. In this work, the belonging of an $e \in \mathcal{I}_{\Psi,\xi}$ to the preconditions, positive effects or negative effects of $\xi$ is handled with a refined propositional encoding that uses fluents of two types, $pre\_\xi\_e$ and $eff\_\xi\_e$, instead of the three fluents used in the BLS. The four possible combinations of these two fluents are sumarized in Figure 1. This compact encoding allows for a more effective exploitation of the syntactic constraints, and also yields the solution space of $\Lambda$ to be the same as its search space.

## The sampling space

The single plan observation of $\mathcal{O}$ is defined as $\mathcal{O} = \langle s_0^o, s_1^o \ldots, s_m^o \rangle$, a sequence of possibly *partially observed states* except for the initial state $s_0^o$ which is a *fully observable* state. As commented before, the predicates $\Psi$ and the objects that shape the fluents $F$ are then deducible from $s_0^o$. A partially observed state $s_i^o$, $1 \leq i \leq m$, is one in which $|s_i^o| < |F|$; i.e., a state in which at least a fluent of $F$ was not observed. Intermediate states can be *missing*, meaning that they are unobservable, so transiting between two consecutive observed states in $\mathcal{O}$ may require the execution of more than one action ($\theta(s_i^o, \langle a_1, \ldots, a_k \rangle) = s_{i+1}^o$ (with $k \geq 1$ is unknown but finite). The minimal expression of a learning sample must comprise at least two state observations, a full initial state $s_0^o$ and a partially observed final state $s_m^o$ so $m \geq 1$.

Figure 2 shows a learning example that contains an initial state of the blocksworld where the robot hand is empty and three blocks (namely blockA, blockB and blockC) are on top of the table and clear. The observation represents a partially observable final state in which blockA is on top of blockB and blockB on top of blockC.

```
(:predicates (on ?x ?y) (ontable ?x)
      (clear ?x) (handempty)
      (holding ?x))

(:objects blockA blockB blockC)

(:init (ontable blockA) (clear blockA)
      (ontable blockB) (clear blockB)
      (ontable blockC) (clear blockC)
      (handempty))

(:observation (on blockA blockB) (on blockB blockC))
```

Figure 2: Example of a two-state observationn for the learning STRIPS action models.

## The domain-specific knowledge

Our approach is to introduce *domain-specific knowledge* in the form of *state constraints* to further restrict the space of the action models. Back to the *blocksworld* domain, one can argue that on($v_1$,$v_1$) and on($v_2$,$v_2$) will not appear in the $pre(\xi)$, $del(\xi)$ and $add(\xi)$ of any action model $\xi$ because, in this specific domain, a block cannot be on top of itself. The notion of state constraint is very general and has been used in different areas of AI and for different purposes. In planning, state constraints are compact and abstract representations that relate the values of variables in each state traversed by a plan, and allow to specify the set of states where a given action is applicable, the set of states where a given *axiom* or *derived predicate* holds or the set of states that are considered goal states (Haslum *et al.* 2018).

*State invariants* is a useful type of state constraints for computing more compact state representations of a given planning problem (Helmert 2009) and for making *satisfiability planning* or *backward search* more efficient (Rintanen 2014; Alcázar and Torralba 2015). Given a planning problem $P = \langle F, A, I, G \rangle$, a state invariant is a formula $\phi$ that holds in $I$, $I \models \phi$, and in every state $s$ built out of $F$ that is reachable by applying actions of $A$ in $I$.

A *mutex* (mutually exclusive) is a state invariant that takes the form of a binary clause and indicates a pair of different properties that cannot be simultaneously true (Kautz and Selman 1999). For instance in a three-block *blocksworld*, $\neg on(block_A, block_B) \vee \neg on(block_A, block_C)$ is a *mutex* because $block_A$ can only be on top of a single block.

Recently, some works point at extracting *lifted* invariants, also called *schematic* invariants (Rintanen 2017), that hold for any possible state and any possible set of objects. Invariant templates obtained by inspecting the lifted representation of the domain have also been exploited for deriving *lifted mutex* (Bernardini *et al.* 2018). In this work we exploit domain-specific knowledge that is given as *schematic mutex*. We pay special attention to *schematic mutex* because they identify mutually exclusive properties of a given type of objects (Fox and Long 1998) and because they enable (1) an effective completion of a partially observed state and (2) an effective pruning of inconsistent STRIPS action models.

We define a schematic mutex as a $\langle p, q \rangle$ pair where both $p, q \in \mathcal{I}_{\xi,\Psi}$ are predicates that shape the preconditions or effects of a given action scheme $\xi$ and they satisfy the formulae $\neg p \vee \neg q$, considering that their corresponding variables are universally quantified. For instance, $holding(v_1)$ and $clear(v_1)$ from the *blocksworld* are *schematic mutex* while $clear(v_1)$ and $ontable(v_1)$ are not because $\forall v_1, \neg clear(v_1) \vee \neg ontable(v_1)$ does not hold for every possible state. Figure 3 shows an example of four clauses that define schematic mutexes for the *blocksworld* domain.

$\forall x_1, x_2 \ \neg ontable(x_1) \vee \neg on(x_1, x_2)$.
$\forall x_1, x_2 \ \neg clear(x_1) \vee \neg on(x_2, x_1)$.
$\forall x_1, x_2, x_3 \ \neg on(x_1, x_2) \vee \neg on(x_1, x_3)$ *such that* $x_2 \neq x_3$.
$\forall x_1, x_2, x_3 \ \neg on(x_2, x_1) \vee \neg on(x_3, x_1)$ *such that* $x_2 \neq x_3$.

Figure 3: *Schematic mutexes* for the *blocksworld* domain.

## Action model learning from *schematic mutexes*

In this section, we show how to exploit *schematic mutexes* for solving the learning task $\Lambda = \langle \mathcal{M}, \mathcal{O}, \Phi \rangle$.

## Completing partially observed states with *schematic mutexes*

The addition of new literals to complete the partial states $\langle s_1^o \ldots, s_m^o \rangle$ of an observation $\mathcal{O}$ using a set of schematic mutexes $\Phi$ is done in a pre-processing stage.

Let $\Omega$ be the set of objects that appear in $F$ as the values of the arguments of the predicates $\Psi$, and $\phi = \langle p, q \rangle$ a schematic mutex. There exist many possible instantiations of $\phi$ of the type $\langle p(\omega), q(\omega') \rangle$ with objects of $\Omega$, where $\omega \subseteq \Omega^{|args(p)|}$ and $\omega' \subseteq \Omega^{|args(q)|}$. Let us now assume that the instantiation $p(\omega) \in s_j^o$, $(1 \le j \le m)$, being $s_j^o$ a partially observed state of $\mathcal{O}$. Then, two situations may occur: (a) $\neg q(\omega') \in s_j^o$, in which case the expression $\neg p(\omega) \vee \neg q(\omega')$ holds in $s_j^o$; or (b) $\neg q(\omega') \notin s_j^o$, in which case the literal has not been observed in $s_j^o$ and so we can safely complete the state with $\neg q(\omega')$ (the same applies inversely, when $q(\omega') \in s_j^o$ but $\neg p(\omega) \notin s_j^o$). In other words, if we find that one component of a schematic mutex is positively observed in a state and the other component is not observable in such state, we can complete the state with the missing negative

| ID | Action | New conditional effect |
|---|---|---|
| 1 | $(\texttt{insert\_pre})_{\xi,\texttt{p}}$ | $\{pre\_\xi\_q\} \triangleright \{invalid\}$ |
| 2 | $(\texttt{insert\_eff})_{\xi,\texttt{p}}$ | $\{pre\_\xi\_q \wedge eff\_\xi\_q \wedge pre\_\xi\_p\} \triangleright \{invalid\}$ |
| 3 | $(\texttt{insert\_eff})_{\xi,\texttt{p}}$ | $\{\neg pre\_\xi\_q \wedge eff\_\xi\_q \wedge \neg pre\_\xi\_p\} \triangleright \{invalid\}$ |
| 4 | $(\texttt{apply})_{\xi,\omega}$ | $\{\neg pre\_\xi\_p \wedge eff\_\xi\_p \wedge$ $q(\omega) \wedge \neg pre\_\xi\_q\} \triangleright \{invalid\}$ |
| 5 | $(\texttt{apply})_{\xi,\omega}$ | $\{\neg pre\_\xi\_p \wedge eff\_\xi\_p \wedge$ $q(\omega) \wedge \neg eff\_\xi\_q\} \triangleright \{invalid\}$ |

Figure 4: Summary of the new conditional effects added to the classical planning compilation for the learning of STRIPS action models.

literal. For instance, if the literal `holding(blockA)` is observed in a particular state and $\Phi$ contains the schematic mutex $\neg holding(v_1) \vee \neg clear(v_1)$, we extend the state observation with literal $\neg$`clear(blockA)` (despite this particular literal being initially unknown).

## Pruning inconsistent action models with *schematic mutexes*

Our approach to learning action models consistent with the schematic mutexes in $\Phi$ is to ensure that newly generated states induced by the learned actions do not introduce any inconsistency. This is implemented by adding new conditional effects to the `insert` and `apply` actions of the BLS compilation. Figure 4 summarizes the new conditional effects added to the compilation and next, we describe them in detail:

1-3 For every schematic mutex $\langle p, q \rangle$, where both $p$ and $q$ belong to $\mathcal{I}_{\xi,\Psi}$, one conditional effect is added to the $(\texttt{insert\_pre})_{\xi,\texttt{p}}$ actions to prevent the insertion of two preconditions that are schematic mutex. Likewise, two conditional effects are added to the $(\texttt{insert\_eff})_{\xi,\texttt{p}}$ actions, one to prevent the insertion of two positive effects that are schematic mutex and another one to prevent two mutex negative effects.

4-5 For every schematic mutex $\langle p, q \rangle$, where both $p$ and $q$ belong to $\mathcal{I}_{\xi,\Psi}$, two conditional effects are added to the $(\texttt{apply})_{\xi,\omega}$ actions to prevent positive effects that are inconsistent with an input observation (in $(\texttt{apply})_{\xi,\omega}$ actions the variables in $pars(\xi)$ are bounded to the objects in $\omega$ that appear in the same position).

In theory, conditional effects of the type 4-5 are sufficient to guarantee that all the states traversed by a plan produced by the compilation are *consistent* with the input set of schematic mutexes $\Phi$ (obviously provided that the input initial state $s_0^o$ is a valid state). In practice we include also conditional effects of the type 1-3 because they prune *invalid* action models at an earlier stage of the planning process (these effects extend the `insert` actions that always appear first in the solution plans).

## Compilation properties

**Lemma 1.** *Soundness. Any classical plan $\pi$ that solves $P'$ (planning task that results from the compilation) produces a model $\mathcal{M}'$ that solves the $\Lambda = \langle \mathcal{M}, \mathcal{O}, \Phi \rangle$ learning task.*

*Proof.* According to the $P'$ compilation, once a given precondition or effect is inserted into the domain model $\mathcal{M}$ it cannot be undone. In addition, once an action model is applied it cannot be modified. In the compiled planning task $P'$, only $(\texttt{apply})_{\xi,\omega}$ actions can update the value of the state fluents $F$. This means that a state consistent with an observation $s_m^o$ can only be achieved executing an applicable sequence of $(\texttt{apply})_{\xi,\omega}$ actions that, starting in the corresponding initial state $s_0^o$, validates that every generated intermediate state $s_j$ $(0 < j \le m)$, is consistent with the input state observations and *state-invariants*. This is exactly the definition of the solution condition for model $\mathcal{M}'$ to solve the $\Lambda = \langle \mathcal{M}, \mathcal{O}, \Phi \rangle$ learning task. $\qquad \square$

**Lemma 2.** *Completeness. Any model $\mathcal{M}'$ that solves the $\Lambda = \langle \mathcal{M}, \mathcal{O}, \Phi \rangle$ learning task can be computed with a classical plan $\pi$ that solves $P'$.*

*Proof.* By definition $\mathcal{I}_{\xi,\Psi}$ fully captures the set of elements that can appear in an action model $\xi$ using predicates $\Psi$. In addition the $P'$ compilation does not discard any model $\mathcal{M}'$ definable within $\mathcal{I}_{\xi,\Psi}$ that satisfies the mutexes in $\Phi$. This means that, for every model $\mathcal{M}'$ that solves the $\Lambda = \langle \mathcal{M}, \mathcal{O}, \Phi \rangle$, we can build a plan $\pi$ that solves $P'$ by selecting the appropriate $(\texttt{insert\_pre})_{\xi,e}$ and $(\texttt{insert\_eff})_{\xi,e}$ actions for programming the precondition and effects of the corresponding action models in $\mathcal{M}'$ and then, selecting the corresponding $(\texttt{apply})_{\xi,\omega}$ actions that transform the initial state observation $s_0^o$ into the final state observation $s_m^o$. $\qquad \square$

The size of $P'$ depends on the arity of the predicates in $\Psi$, that shape variables $F$, and the number of parameters of the action models, $|pars(\xi)|$. The larger these arities, the larger $|\mathcal{I}_{\xi,\Psi}|$. The size of $\mathcal{I}_{\xi,\Psi}$ is the most dominant factor of the compilation because it defines the $pre\_\xi\_e/eff\_\xi\_e$ fluents, the corresponding set of $\texttt{insert}$ actions, and the number of conditional effects in the $(\texttt{apply})_{\xi,\omega}$ actions. Note that *typing* can be used straightforward to constrain the FOL interpretations of $\Psi$ over the parameters $pars(\xi)$, which will significantly reduce $|\mathcal{I}_{\xi,\Psi}|$ and hence the size of $P'$ output by the compilation.

Classical planners tend to prefer shorter solution plans, so our compilation (as well as the BLS) may introduce a bias to $\Lambda = \langle \mathcal{M}, \mathcal{O}, \Phi \rangle$ learning tasks preferring solutions that are referred to action models with a shorter number of preconditions/effects. In more detail, all $\{pre\_\xi\_e, eff\_\xi\_e\}_{\forall e \in \mathcal{I}_{\xi,\Psi}}$ fluents are false at the initial state of our $P'$ compilation so classical planners tend to solve $P'$ with plans that require a smaller number of $\texttt{insert}$ actions.

This bias can be eliminated defining a cost function for the actions in $P'$ (e.g. $\texttt{insert}$ actions have *zero cost* while $(\texttt{apply})_{\xi,\omega}$ actions have a *positive constant cost*). In practice we use a different approach to disregard the cost of $\texttt{insert}$ actions since classical planners are not proficient at optimizing plan cost with zero-cost actions. Instead, our approach is to use a SAT-based planner (Rintanen 2014) that can apply all actions for inserting preconditions in a single planning step (these actions do not interact). Further, the actions for inserting action effects are also applied in another single planning step. The plan horizon for programming any action model is then always bounded to 2. The SAT-based planning approach is also convenient for its ability to deal with planning problems populated with dead-ends and because symmetries in the insertion of preconditions/effects into an action model do not affect the planning performance.

# Evaluation

This section evaluates the improvement when using domain-specific knowledge for learning action models.

**Reproducibility** The domains used in the evaluation are IPC domains that satisfy the STRIPS requirement (Fox and Long 2003), taken from the PLANNING.DOMAINS repository (Muise 2016). For each domain we generated 10 learning examples of length 10 via random walks and report average values (only a single example is considered at each learning episode). We also introduce a new parameter, the *degree of observability* $\sigma$, which indicates de probability of observing a literal in an intermediate state. This parameter is used to build observations with varying degrees of incompleteness. All experiments are run on an Intel Core i5 3.10 GHz x 4 with 16 GB of RAM.

For the sake of reproducibility, the compilation source code, evaluation scripts, used benchmarks and input *state-invariants* are fully available at the repository *https://github.com/anonsub/oneshot-learning*.

**Metrics** The learned models are evaluated using the *precision* and *recall* metrics for action models proposed in (Aineto *et al.* 2018), which compare the learned models against the reference model.

Precision measures the correctness of the learned models. Formally, $Precision = \frac{tp}{tp+fp}$, where $tp$ is the number of true positives (predicates that appear in both the learned and reference action models) and $fp$ is the number of false positives (predicates that appear in the learned action model but not in the reference model). Recall, on the other hand, measures the completeness of the model and is formally defined as $Recall = \frac{tp}{tp+fn}$ where $fn$ is the number of false negatives (predicates that should appear in the learned action model but are missing).

## Observability versus Knowledge

In our first experiment, we seek to answer the question as to whether the plan observation (single learning example) of $\mathcal{O}$ is replaceable by the domain knowledge encoded in $\Phi$. To this end, we evaluate the following 4 settings:

1. **Minimal observability:** This is the baseline setting where we use the minimal expression of the single learning example; i.e., a fully observed initial state and a partially observed final state. This setting is labeled as $\sigma = 0$ with no $\Phi$.

2. **Only knowledge ($\Phi$):** In this setting we add domain knowledge encoded as schematic mutexes to the baseline scenario ($\sigma = 0$ with $\Phi$).

3. **Only observability:** We use a more complete observation where intermediate states of the learning example are partially observed ($\sigma = 0.2$ with no $\Phi$).

4. **Both observability and $\Phi$:** We use both a more complete observation and schematic mutexes ($\sigma = 0.2$ with $\Phi$).

| | $|\Phi|$ | setting 1 $\sigma = 0$ w/o $\Phi$ | | setting 2 $\sigma = 0$ with $\Phi$ | | setting 3 $\sigma = 0.2$ w/o $\Phi$ | | setting 4 $\sigma = 0.2$ with $\Phi$ | |
|---|---|---|---|---|---|---|---|---|---|
| | | P | R | P | R | P | R | P | R |
| blocks | 9 | 0.51 | 0.36 | 0.54 | 0.23 | 0.59 | 0.49 | 0.79 | 0.70 |
| driverlog | 8 | 0.48 | 0.36 | 0.34 | 0.34 | 0.41 | 0.31 | 0.69 | 0.49 |
| ferry | 2 | 0.47 | 0.39 | 0.58 | 0.43 | 0.50 | 0.51 | 0.61 | 0.72 |
| floor-tile | 7 | 0.39 | 0.39 | 0.48 | 0.45 | 0.64 | 0.48 | 0.74 | 0.52 |
| grid | 3 | 0.40 | 0.31 | 0.42 | 0.31 | 0.43 | 0.31 | 0.50 | 0.37 |
| gripper | 5 | 0.76 | 0.50 | 0.77 | 0.51 | 0.85 | 0.74 | 0.92 | 0.81 |
| hanoi | 3 | 0.88 | 0.71 | 0.73 | 0.81 | 0.94 | 0.78 | 1.00 | 0.81 |
| n-puzzle | 3 | 0.94 | 0.76 | 0.95 | 0.81 | 0.97 | 0.86 | 0.97 | 0.89 |
| parking | 8 | 0.54 | 0.41 | 0.60 | 0.40 | 0.51 | 0.40 | 0.51 | 0.41 |
| transport | 4 | 0.45 | 0.45 | 0.53 | 0.46 | 0.49 | 0.37 | 0.94 | 0.75 |
| zeno-travel | 4 | 0.73 | 0.36 | 0.80 | 0.36 | 0.79 | 0.36 | 0.89 | 0.50 |
| | | 0.60 | 0.45 | 0.61 | 0.46 | 0.65 | 0.51 | 0.78 | 0.63 |

Table 1: Observability versus knowledge

Table 1 shows the average values of precision (P) and recall (R) for each domain in the four tested settings. The table also reports the number of schematic mutexes ($|\Phi|$) used for each domain. Comparing the settings *only domain knowledge* (setting 2) with *only observability* (setting 3), we can see that slightly better results are obtained with the latter, meaning that observability is more informative than the used domain knowledge. On the other hand, the gain of using $\Phi$ under minimal observability (setting 1 compared to setting 2) is rather marginal. While these results might indicate a general preference for observations over knowledge, when comparing setting 3 with setting 4, we can observe a significant improvement in the quality of the learned models. This indicates that the payoff of using $\Phi$ is noticeable when the learning example has a certain degree of observability.

### Using knowledge to counter incompleteness

The previous experiment reveals that observations are not totally replaceable by domain knowledge; but also shows that given a minimum degree of observability, using $\Phi$ enriches both the observations and the learning process and better models are learnable. In this next experiment we measure the improvement provided by $\Phi$ at increasing degrees of observability of the learning example.

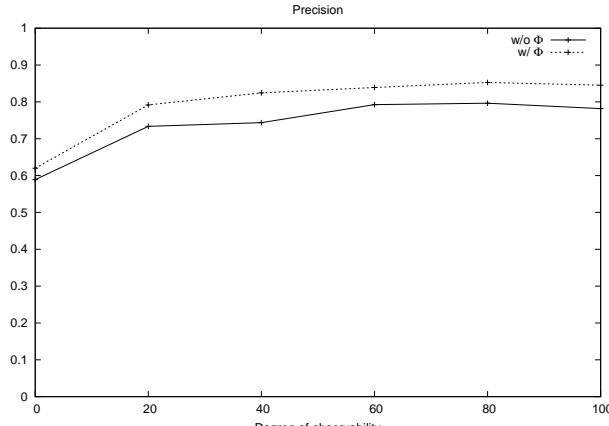

Figure 5: Comparison of the precision of the learned models for increasing degrees of observability.

Figures 5 and 6 compare the Precision and Recall of the

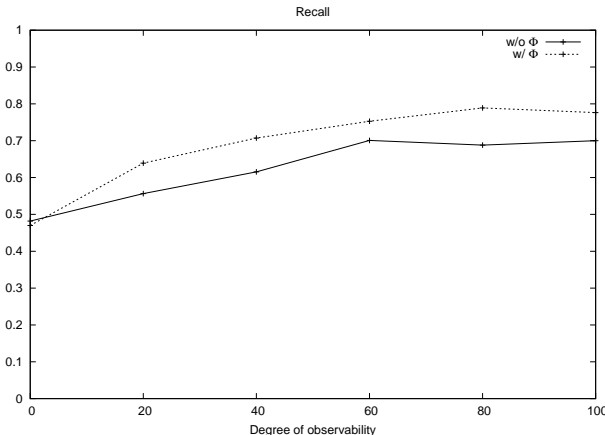

Figure 6: Comparison of the recall of the learned models for increasing degrees of observability.

learned models with and without domain knowledge. The points plotted in these figures are average values across all the domains presented in Table 1. The results show that using $\Phi$ significantly improves the learned models no matter how complete the learning examples are. An interesting and revealing aspect from the figures is that the quality of the action models learned with 30%-observable learning examples and $\Phi$ is comparable to the quality obtained with a 100%-observable example. Hence, domain knowledge can make up for the lack of completeness in the learning examples.

## Conclusions

We present an approach to learn action models that builds upon a former compilation-to-planning learning system (Aineto *et al.* 2018). Our proposal studies the gains of using domain-specific knowledge when the availability (amount and observability) of learning examples is very limited. Introducing domain knowledge encoded as schematic mutexes allows to narrow down the search space of the learning task and improve overall the performance of the learning system to the point that it offsets the lack of learning examples.

In a theoretical work that analyzes the relation between the number of observed trajectory plans and the guarantee for a learned action model to achieve the goal (Stern and Juba 2017), authors conclude that the number of trajectories needed scales gracefully and the guarantee grows linearly with the number of predicates and quasi-linearly with the number of actions. This evidences that learning accurate models is heavily dependent on the number and quality (observability) of the learning examples. In this sense, our proposal comes to alleviate this dependency by relying on easily deducible domain knowledge. It is not only capable of learning from a single non-fully observable learning example but also proves that learning from a 30%-observable example with domain-specific knowledge is comparable to learning from a complete plan observation.

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
