# OpenReview forum: "One-shot learning: From domain knowledge to action models"
_icaps-conference.org/ICAPS/2019/Workshop/KEPS — KEPS 2019_

### Official Review · AnonReviewer1 · 2019-05-09
**Novel idea for action-model learning that needs more broader justification**

**Rating:** 4
**Confidence:** 3

**Review:**

The paper is about learning action models by exploiting observations of states for executed plans and also using state axioms providing additional knowledge about the domain. The paper is clearly relevant to KEPS and there is novel though a bit incremental contribution. State axioms are known from planning for some time and the authors suggest using their specific version - mutexes (called schematic mutexes in the paper) - in action-model learning using compilation to planning. The preliminary experiments show that it brings some advantage over the “classical” approach, in particular when some intermediate states are observed. I think it is an interesting approach as it goes in the direction of exploiting more knowledge, that is frequently available, rather than using brute force with a large number of training data (which is not always available).

Some specific comments:

The idea of using state axioms can be applied in other action-model learning approaches, correct? Specifically, it seems that ARMS is ready for this approach as the domain constraints can be directly encoded in the SAT formula. Maybe, it would be even easier to encode the constraints there than to do it in PDDL (see the next comment).

The authors show how to encode schematic mutexes for the compilation-based approach. It is not fully clear how complicated encoding other types of state axioms will be. Some discussion of generality of the proposed approach would be useful.

As I understand some of the mutex constraints are resolved in preprocessing while others are encoded in the compiled model. It is enough if everything is encoded in the compiled model so resolving some mutexes in pre-processing only speeds-up the process, correct? If yes, the empirical results can also demonstrate how much it helps.

Why actions are not assumed among observations? The compilation-based approach supports partially observed states and actions. Is there any specific reason to omit actions in this setting?

The section on background uses a bit non-standard model of planning, where the state is described as a complete set of valid literals (so predicates that are not true in the state are explicitly included in a negated version). Then, one must be careful in defining the transition function, specifically, the complement of effects. This is not a set complement but a logical complement, which is not fully clear from the text. Also, using conditional effects brings a danger if different conditional effects are contradictory. Then, based on the included definition, we may obtain an inconsistent state as it may contain some predicate in both positive and negative version (from two different effects with satisfied conditions).

When introducing schematic mutexes to the compilation approach (Figure 4), it looks like they are used to generate a failure during planning. Would not be better to prevent such a failure rather than just wait for it? The whole compilation-based approach resembles the generate-and-test approach (generate the model and then verify it) so it would be interesting to look for something more efficient. Actually, runtimes are never reported in the paper so we do not know how practically efficient the approach is.

The system is using a SAT-based planner, which removes many obvious symmetries when generating the action model. The authors claim in the text that the planning horizon is bound to 2 (for programming any action model). It is a correct claim (though might be hard to grasp by the reader who is not aware of details of the compilation approach), but it may be also a bit misleading as we need additional layers to encode the observed plan (yes, this is not for programming the model but for evaluating it, but we need more layers). So I do not fully understand the discussion on bias to find shorter plans as the bias is still there (the systems tries to minimise the number of actions that explain given observations).

The authors use a good number of domains in experiments. For each domain, they define some set of schematic mutexes, but the reader does not know how they look. It is expected, that quality of this extra domain knowledge will influence efficiency so the empirical results are always biased by what extra knowledge is provided. As the extra knowledge has a compact form (set of mutexes), it might be useful to study how the results are influenced by enlarging this set. Right now the comparison is between zero and all knowledge.

---

### Official Review · AnonReviewer2 · 2019-05-11
**A clean investigation of supporting action model learning by providing the complete set of mutex relations**

**Rating:** 4
**Confidence:** 2

**Review:**

The paper describes a domain acquisition approach that extends (Aineto et al. 2018). The underlying work takes partially observed plan sequences and a partial domain model (possibly empty) as input and creates a domain model through a construction of a planning problem. The presented work considers using a different representation for the partial domain model. In particular, using schematic mutex relations in place of pre/post conditions. The approach exploits the mutexes in two ways. Firstly attempting to fill in the partially observed state information. Secondly in order to shape the chosen domain model.

In the context of domain model acquisition the direction of this work is refreshing. It is typical in this area to move between data inputs with different properties (even subtly), leaving meaningful analysis difficult or meaningless. This work fits in nicely as part of a more systematic investigation of how different input types can be exploited.

The paper is well presented and the results indicate that schematic mutexes provide considerable leverage for learning.

A key consideration here is to what extent these mutexes are easier to specify than the domain model itself, which includes whether they are conceptually simpler than specifying the domain mechanics. I don't believe this point is really addressed (other than claiming it as "easily deducible domain knowledge"). This being the important topic of the paper, it seems there is interest in exploring it in the evaluation. For example, what happens if only x% of the mutexes are specified?

The other aspect that would be interesting to understand better is the scalability of this approach. Given the modest size of typical benchmark domain models and your use of 16GB RAM in the experiments, this is left unclear.

I am surprised by the related work section, which appears to focus on domain model acquisition approaches that deal with noise. It is hardly surprising that these approaches require more evidence. Similarly it is unclear to what extent this new approach really brings anything new to general properties of completeness and correctness with respect to the acquired model.